# Asymmetry Thresholds Reflecting the Visual Assessment of Forelimb Lameness on Circles on a Hard Surface

**DOI:** 10.3390/ani13213319

**Published:** 2023-10-25

**Authors:** Claire Macaire, Sandrine Hanne-Poujade, Emeline De Azevedo, Jean-Marie Denoix, Virginie Coudry, Sandrine Jacquet, Lélia Bertoni, Amélie Tallaj, Fabrice Audigié, Chloé Hatrisse, Camille Hébert, Pauline Martin, Frédéric Marin, Henry Chateau

**Affiliations:** 1Labcom LIM-ENVA, LIM France, 24300 Nontron, France; claire.macaire@vet-alfort.fr (C.M.);; 2CIRALE, USC 957 BPLC, Ecole Nationale Vétérinaire d’Alfort, 94700 Maisons-Alfort, France; 3Laboratoire de BioMécanique et BioIngénierie (UMR CNRS 7338), Centre of Excellence for Human and Animal Movement Biomechanics (CoEMoB), Université de Technologie de Compiègne (UTC), Alliance Sorbonne Université, 60200 Compiègne, France

**Keywords:** horse, lameness, symmetry, receiver operating characteristic curves, inertial measurement unit, circle

## Abstract

**Simple Summary:**

Veterinary lameness examination commonly involves a visual evaluation of a horse trotting on a circle. Lameness detection can be aided by objective gait analysis, which is used to quantify the movement asymmetry of horses. However, the asymmetry thresholds defined for the trot on a straight line are not applicable to the circles because turning induces physiological asymmetric movement. Four Asymmetry Indices (AIs) were calculated to compare the vertical movement of the head and of the withers between the right limb movement and the left limb movement during a trot stride. This study aims to select the AIs with good discriminative power between a group of sound horses and a group of horses showing consistent unilateral lameness (grade > 1/10) across both circle directions (clockwise, counter clockwise) on a hard surface, and to define the optimal threshold value, based on sensitivity and specificity. Head vertical movement asymmetry showed the highest sensitivity and specificity to detect forelimb lameness when the lame limb was on the inside of the circle, while withers vertical movement asymmetry showed the highest sensitivity and specificity to detect forelimb lameness when the lame limb was on the outside of the circle.

**Abstract:**

The assessment of lameness in horses can be aided by objective gait analysis tools. Despite their key role of evaluating a horse at trot on a circle, asymmetry thresholds have not been determined for differentiating between sound and lame gait during this exercise. These thresholds are essential to distinguish physiological asymmetry linked to the circle from pathological asymmetry linked to lameness. This study aims to determine the Asymmetry Indices (AIs) with the highest power to discriminate between a group of sound horses and a group of horses with consistent unilateral lameness across both circle directions, as categorized by visual lameness assessment conducted by specialist veterinarians. Then, thresholds were defined for the best performing AIs, based on the optimal sensitivity and specificity. AIs were calculated as the relative comparison between left and right minima, maxima, time between maxima and upward amplitudes of the vertical displacement of the head and the withers. Except the AI of maxima difference, the head AI showed the highest sensitivity (≥69%) and the highest specificity (≥81%) for inside forelimb lameness detection and the withers AI showed the highest sensitivity (≥72%) and the highest specificity (≥77%) for outside forelimb lameness detection on circles.

## 1. Introduction

The evaluation of equine locomotion during a trot on a circle represents an essential component of the veterinary lameness assessment [1]. Indeed, the need to produce centripetal force on a circle and the resulting body lean angle induce specific forces on the limb and 3D joint movements compared to the straight line, like a higher loading force on the outside limb, and collateromotion and axial rotation in the digital joints [1,2]. These specific biomechanical constraints on the anatomical structures are considered to be responsible for a different symptomatology between the circle and the straight line [2]. Therefore, this condition provides key information to aid clinical decision making [1].

Nowadays, the assessment of lameness can be aided by objective gait analysis tools [3,4,5,6]. Essential for the clinical application of these tools, asymmetry thresholds have been defined to distinguish between non-lame and lame gait [7,8,9,10,11,12]. McCracken et al. [13] have determined a threshold of 6 mm for the difference in the vertical displacement of the head between the right stance phase and left stance phase for defining forelimb lameness on a straight line. Pfau et al. [14] have established another value for this threshold of 14.5 mm using another objective gait analysis system and in the specific context of screening Thoroughbreds in race training. Lastly, asymmetry thresholds for indices normalized with the Range Of Movement (ROM) have been determined for discriminating visually assessed forelimb lameness on the straight line [15]. In this study, it was shown that the asymmetry of the upward movement of the withers had the highest power to discriminate between sound and forelimb-lame horses. The associated thresholds were −7% asymmetry for left forelimb lameness and +10% asymmetry for right forelimb lameness. These thresholds were expressed in percentage contrary to the previous thresholds expressed in millimeters [13,14], as relative values can be seen as a better way of facilitating comparisons between horses of different sizes and between objective gait analysis systems with different signal processing [16]. Moreover, time-related indices of head movement have proven to be affected by forelimb lameness on a straight line [17,18,19].

Asymmetry thresholds for lameness detection determined on a straight line are not applicable on a circle [20]. Yet, their usefulness in this circumstance of locomotion is all the more crucial as a large proportion of the movements recorded on the circle must be considered as physiologically asymmetric [11,12,20,21,22]. It is therefore essential to highlight the boundary between the physiological asymmetric movement of the circle and an excessive asymmetric movement which should be considered pathological. Asymmetries measured in sound horses during lunging at trot have shown variation higher than previously defined threshold values [11,12,20,21,22]. This variation could be due to factors such as speed, radius and body lean angle, which have been identified as impacting the measurements [23,24].

Thresholds for lameness detection on circles would help the interpretation of asymmetry values provided by an objective gait analysis tool during the entire locomotor examination. The objectives of this study were (i) to select asymmetry indices with the highest sensitivity and specificity to reflect the visual assessment of a veterinary specialist and then (ii) to define the optimal asymmetry thresholds for the inside and outside forelimb lameness detection on circles on a hard surface.

## 2. Materials and Methods

This clinical observational retrospective study was approved by the Clinical Research Ethics Committee (ComERC n◦2022-01-19).

### 2.1. Horses and Conditions

This study was conducted on horses that presented at the Equine Clinic (CIRALE) of the National Veterinary College of Alfort (Maisons-Alfort, France) for locomotor evaluation from April 2019 to February 2023. After collecting the anamnesis and performing the inspection and the palpation of the locomotor system, the veterinarian evaluated the horse locomotion without warm-up. As part of the dynamic locomotor examination, horses were trotted by their owner/groom on the lunge on a circle of 8–14 m diameter, on the left rein (counter clockwise direction) and then on the right rein (clockwise direction). The ground surface was made of hard rubber pavers. Visual evaluation was performed by one of the five veterinary specialists who graduated as DESV (French certification as a specialist in equine locomotor pathology) and were certified by the ISELP (International Society of Equine Locomotor Pathology).

In total, 574 horses were screened and were visually evaluated at lunge on the left rein and on the right rein on a hard surface. Among them, 95 horses showed left forelimb lameness (LF) and 122 horses showed right forelimb lameness (RF) on both reins. Horses showing a lameness grade of 1/10 or over 7/10 on a 0–10 grade scale equivalent to the UK scale (where 0 is sound and 10 is non-weight bearing lameness) on one rein were excluded [25,26,27]. After exclusion, 61 horses showed LF and 76 horses showed RF grade 2–7 lameness on both reins. Horses showing lameness on multiple limbs on one rein were excluded (*n* = 8). With these criteria, 57 horses showed LF lameness and 70 horses showed RF lameness on both reins. A flowchart is provided as a Appendix A. Lameness grades included in each group on both reins are summarized in Table 1.

Thirty-one horses were included in the group of “sound” horses according to the following criteria: (1) the horses were in training and judged by their owners to be capable of performing all the exercises required for their sport level; (2) three of the five veterinary specialists independently watched blinded videos of the horse at walk and at trot, on a hard circle on both reins and on a hard straight line and did not notice any locomotion abnormalities in the entire video (<1/10 lameness grade).

### 2.2. Data Collection

During the locomotor examination, as part of the clinical routine, horses were systematically equipped with the EQUISYM^®^ system (Arioneo, LIM France, Le Bouscat, France), described by Macaire et al. and Timmerman et al. [15,28]. Data were recorded during approximately 20 s of trot per direction (clockwise/counterclockwise). The trot is of particular interest because a stride is composed of two diagonal beats whether on a straight line or circling to left or right.

### 2.3. Data Processing

The data were processed following the methods described by Macaire et al. [15]. Briefly, based on the vertical displacement of the head (_H) and the withers (_W) occurring along a stride, the following asymmetry indices (AIs), expressed as a percentage of the maximal range of motion within a stride, were used to compare the left vs. right part of the stride (Figure 1): AI-Min, AI-Max and AI-up. Additionally, AI-Tmax was calculated as a time-related index, representing the left–right difference in the duration of the down–up cycle (the time between the two maxima) of the vertical displacement occurring before and after the stance phase. AI-Tmax was expressed as a percentage of the maximal time between two consecutive maxima. A positive AI value indicated a smaller movement amplitude or duration during the right stance than during the left stance, and a negative AI value indicated the opposite.

### 2.4. Data Analysis

Descriptive analyses, including mean and standard deviation (SD), were calculated for each variable, for each group (sound/lame) and for each condition (clockwise/counter-clockwise direction). The four AIs calculated from the head and the withers were analyzed using the same methods as those described by Macaire et al. [15]. Normality was assessed using a graphical method [29]. Receiver Operating Characteristic (ROC) curves were plotted to discriminate, respectively, the right and left forelimb lame group (RF, LF) from the control group (sound horses) at lunge separately on the left and on the right rein. The AIs’ discriminative power is indicated by the Area Under Curve (AUC) of the ROC curve. Given that AUC < 50% indicates discrimination no better than chance, higher AUC values (closer to 100%) indicate higher discriminative power. Finally, thresholds of indices with good discriminative power were calculated. In this study, indices were considered as having good discriminative power if the sum of sensitivity and specificity was strictly higher than 150% [30].

## 3. Results

### 3.1. Descriptive Results

The 158 horses included were 78 geldings, 72 females and 8 stallions; 94 Selle Français, 8 Zangersheide, 7 Hanoverian, 6 Koninklijk Warmbloed Paardenstamboek Nederland (KWPN), 6 French riding pony and 37 other breeds; 113 showjumpers, 12 dressage, 10 eventing and 23 other disciplines; and they were aged from 3 to 20 years (mean ± SD, 9 ± 3 years). Age, gender, breed and disciplines are detailed for each horse in the Appendix A). A mean ± SD of 20.4 ± 7.2 trot strides, 22.9 ± 5.8 on the right rein and 17.9 ± 7.6 on the left rein, were processed for each recording. The stride duration was 0.79 ± 0.05 s on both reins, respectively, and 0.80 ± 0.04 s for sound horses, 0.79 ± 0.04 s for RF lame horses and 0.78 ± 0.05 s for LF lame horses. The AIs of the head, the withers and the pelvis are detailed for each included horse in the Appendix A shows AIs and ROM values measured on the left rein and Appendix A shows AIs and ROM values measured on the right rein.

The mean ± SD for each AI and for each horse group are summarized in Table 2 and boxplots are displayed in Figure 2. In sound horses, AIs calculated from the head (AI-min_H, AI-max_H and AI-up_H) were negative on the left rein and positive on the right rein, reflecting a reduced vertical range of motion of the head during the inside forelimb stance phase. AI-min of the withers also showed a reduced minimum (AI-min_W) during the inside forelimb stance phase. On the contrary, AI-max of the withers was positive on the left rein and negative on the right rein, reflecting a reduced maximum after the outside forelimb stance phase. Finally, in sound horses, the AI-Tmax of the head and of the withers, and AI-up of the withers were close to 0% of asymmetry on both reins.

RF-lame horses showed higher mean values (sign of reduced movement on the right) than sound horses for all AIs of the head and withers. On the opposite hand, LF-lame horses showed lower mean values than sound horses for all AIs of the head and withers.

Horses with RF and LF lameness showed, respectively, positive and negative mean values (sign of reduced movement during the stance of the lame limb) for all AIs, except AI-max_W, when the lame limb was on the inside of the circle (RF at right rein and LF at left rein). When the lame limb was on the outside of the circle, AI-min_W was close to 0% of asymmetry.

### 3.2. Discrimination between Lame Horses with the Lame Forelimb on the Inside of the Circle and Sound Horses

ROC curves and associated results for the discrimination of forelimb lameness when the lame limb was on the inside of the circle (RF lameness on right rein and LF lameness on left rein) from sound horses are presented in Figure 3 and Table 3, respectively. The lowest AUC (lowest discrimination performance) value for the inside lameness detection was shown by the AI-max for the head (AUC ≤ 70%) and by the AI-max for the withers (AUC ≤ 67%). The indices with the highest AUC (AUC ≥ 83%) were in the following descending order: AI-min_H, AI-Tmax_H and AI-up_H for RF lameness on the right rein. For LF lameness on the left rein, the indices with the highest AUC (AUC ≥ 87%) were in the following descending order: AI-Tmax_H, AI-min_H and AI-up_H.

### 3.3. Discrimination between Lame Horses with the Lame Forelimb on the Outside of the Circle and Sound Horses

The ROC curves and associated results for the discrimination of forelimb lameness when the lame limb was on the outside of the circle (LF lameness on right rein and RF lameness on left rein) from sound horses are presented in Figure 3 and in Table 4, respectively. The lowest AUC value for the outside lameness detection on the circle was shown by AI-max for the withers (AUC ≤ 71%) and by AI-max for the head (AUC ≤ 75%). The indices with the highest AUC (AUC ≥ 79%) were in the following descending order, AI-up_W, AI-Tmax_W, AI-up_H and AI-min_W, for LF lameness on the right rein. For RF lameness on the left rein, the indices with the highest AUC (AUC ≥ 81%) were in the following descending order: AI-Tmax_W, AI-up_W, AI-up_H and AI-min_W.

### 3.4. Asymmetry Thresholds of Reliable Indices

For discriminating sound horses from lame horses with the lame limb on the inside of the circle, the indices with high sensitivity and specificity (sum of sensitivity and specificity greater than 150%) were three indices from the head (AI-up_H, AI-Tmax_H and AI-min_H). For discriminating forelimb lameness when the lame limb was on the outside of the circle, the indices with greater sensitivity and specificity were three indices from the withers (AI-up_W, AI-Tmax_W and AI-min_W) and one from the head (AI-up_H).

Figure 4 represents the thresholds and their 95% CI associated with a sum of sensitivity and specificity over 150% for inside (Figure 4a) and outside (Figure 4b) lameness discrimination on both reins.

When comparing the absolute values of left rein and right rein thresholds for lameness discrimination when the lame limb was on the inside of the circle, the maximum difference between right and left rein was 4% of asymmetry for AI-Tmax_H (Figure 4a). When the lame limb was on the outside of the circle, the maximum difference between right and left was 8% of asymmetry for AI-up_W (Figure 4b).

## 4. Discussion

Asymmetry Indices (AIs) measured on sound horses showed a reduced head and withers movement during the inside forelimb stance phase, except for the AI-max of the withers. This goes with a larger downward movement of the trunk and head towards the outside forelimb, confirming the results of previous studies [12,22]. This physiological asymmetry observed on the circle means that we need to define the limit between a physiological asymmetry and an asymmetry that is amplified (or reduced) when a pathological phenomenon is superimposed.

Our study confirms the hypothesis that head and withers vertical displacements are indicators of forelimb lameness on the circle in a specific group of horses showing single-limb lameness on both reins, excluding a lameness grade of 1/10. In addition, the present study reveals that when the lame limb is inside the circle, head movements have the highest discriminative power. Conversely, when the lame limb is outside the circle, withers movements have the highest discriminative power. Practically, the AI-min and AI-Tmax of the head discriminated forelimb lameness when the lame limb was on the inside of the circle with the highest sensitivity (≥79%) and specificity (≥81%) on both reins. AI-up and AI-Tmax of the withers discriminated forelimb lameness when the lame limb was on the outside of the circle with the highest sensitivity (≥83% and ≥74%, respectively) and specificity (≥77% and ≥84%, respectively) on both reins.

Among the four indices (AI-min, AI-max, AI-up and AI-Tmax) used in this study, AI-max has systematically the lowest sensitivity and specificity for discriminating horses showing forelimb lameness on both reins from sound horses on a circle. In contrast, a previous study measured a reduced maximum altitude of the head after outside forelimb lameness induction [21]. There are several possible explanations for this discrepancy. On the one hand, the reference population for the two studies is not identical since we chose to concentrate on the lameness visible on both clockwise and counter clockwise circles. On the other hand, the type of lameness analyzed in the study by Rhodin et al. [21] was induced lameness (with a modified horseshoe) and not spontaneous lameness as in the present study.

Several studies describe variation in asymmetry but did not reach a consensus on the physiological asymmetry of sound horses induced by the circle [11,12,21,22]. Rhodin et al. [21] concluded a reduced head movement for the outside forelimb (HDmin). In 2016, Rhodin et al. [22] highlighted a non-uniform effect of the circle on the head. Other studies [11,12] showed results similar to those of the present study concerning physiological head asymmetry on a circle with a reduced head movement during the inside forelimb stance. The recorded asymmetries of the withers vertical displacements on sound horses trotting on a circle are consistent with a previous study [12] and showed a lowered minimum during and a reduced maximum after the stance phase of the outside forelimb. However, in the studies of Starke et al. [12] and Rhodin et al. [22], the inclusion criteria were based on the straight line and did not exclude visible lameness on the circle, meaning that some of the recorded asymmetries in their studies could be due to the apparition of a lameness on the circle that was not visible on the straight line.

Studies agree that forelimb lameness induces a reduced downward movement of the head (AI-min) during the stance phase of the lame limb [10,11,21,31], particularly for the inside lame limb when trotting on a circle [11,21,32]. The importance of the head movements for detecting the inside lameness and the withers for the outside lameness, as observed here, has been mentioned in the detailed results of a linear discriminant analysis discriminating positive and negative anesthesia realized by Pfau et al. [32]. They revealed that the minimum and upward movement asymmetry of the withers were the features with the most important weight to discriminate the anesthesia effect on the outside limb in circles on a hard surface, respectively, in each canonical discriminate function. Also, the minimum altitude of the head had the most important weight for the inside limb in the first canonical discriminate function. Marunova et al. [33], without the separation of outcomes from the inside and the outside limbs, measured significant differences in the head asymmetry indices and an upward withers asymmetry index before and after positive anesthesia.

No thresholds were previously determined for the locomotion on circles, allowing only comparisons with straight lines. For discriminating sound horses from lame horses with the lame limb on the inside of the circle, thresholds of head amplitude asymmetry (AI-up_H) showed higher values (−49% and +49% for respectively LF and RF lame limb) than thresholds for discriminating sound horses from lame horses established using similar methods on a straight line (−36% and +24% for respectively LF and RF lame limb) [15]. Conversely, thresholds of AI-up_H established for discriminating sound horses from lame horses with the lame limb on the outside of the circle showed lower values (+7% and 0% for respectively LF and RF lame limb) than thresholds for discriminating sound horses from lame horses on a straight line (−36 and +24% for respectively LF and RF lame limb) [15]. This difference underlines the absolute need to take into account the physiological asymmetry induced by the circle on the head movements. In contrast, thresholds of withers amplitude asymmetry (AI-up_W) were less influenced by movement on the circle when the lame limb was on the outside (−5% and +13%, for respectively LF and RF lame limb, on the circle compared to −10% and +7%, for respectively LF and RF lame limb, on a straight line) [15].

The results obtained also demonstrate that, on the circle, the values of certain indices must be considered abnormal when they are equal to 0% (perfect symmetry between right and left). For example, when the index of the minimum withers height (AI-min_W) is equal to zero on a circle, this result indicates lameness on the limb outside the circle. An analysis of the symmetry indices on the circle must therefore take account of this shift in relation to what is expected on the straight line. This information is also useful for the clinician, who should consider that perfect symmetry on the circle could be interpreted as suspicious, as physiological asymmetry is to be expected.

In our study population, it appeared that the average grades attributed by the veterinary specialist to the inside forelimb lameness were slightly higher than the grade attributed to outside forelimb lameness. Means of lameness grade were indeed 3.5/10 for the inside forelimb and 3.1/10 for the outside forelimb. One hypothesis could be that injuries affecting the included horses were more painful for the inside limb than for the outside limb [1]. This could be a result of greater collateromotion and axial rotation movements in the inside limb than in the outside limb [1]. But, it should first and foremost be remembered that the main manifestation of an inside lameness is on the head, leading to a possible overestimation of lameness with an obvious head nod compared to a withers asymmetry [34].

In the present study, AIs were divided by the ROM to obtain relative indices expressed as a percentage. ROM values in centimeters are also provided in Appendix A for the locomotion on the right rein and in Appendix A for the locomotion on the left rein, meaning that the absolute value differences between the right and left limbs in centimeters can be easily recalculated using the formula AI(%) × ROM(cm)/100. However, normalizing values seems sensible in order to compare movements measured in a heterogeneous population, including individuals of different sizes (from pony to large horse) with varying vertical amplitudes of gaits. In addition, IMU systems differ in how they process accelerometric signals into displacement and this affects the threshold values that are used across systems [16]. Normalization can facilitate the comparison of asymmetry indices processed by different gait analysis systems, as suggested by Hardeman et al. [35].

In this study, an index based on temporal pattern asymmetry was used. AI-Tmax, reflecting a difference in duration of the down–up cycle between the right and left stance, showed a discriminative power as high as the amplitude asymmetry of the head and the withers. The results showed a reduction in the duration of the down–up cycle (time between two maxima) during the stance phase of the lame limb. Asymmetry indices based on Fourier transform have been shown to be indicators of lameness [17,18,19]. Also, Pfau et al. [36] found that the relative timing of the head movements compared to the withers or the pelvis was also an indicator of lameness. This underlines the importance of these temporal variables in the identification and analysis of lameness.

Horses were included in this study if they were lame on both reins. This selection criterion represents a unique subgroup of lameness cases, showing a specific manifestation. This choice has been made in order to guarantee unambiguous lameness under circle conditions, and in order to study the locomotion of the same horses whether they turn on the right rein or on the left rein. Otherwise, the locomotion of lame horses on the right rein would not be the locomotion of the lame horses, which were studied on the left rein. On the contrary, the group of sound horses would remain unchanged between the two reins, even though they are the reference group for comparison.

A well-known limitation of this study is the clinical reference used to distinguish lame horses from non-lame horses considering that visual clinical assessment is recognized by definition as being subjective [37,38]. This is a deliberate choice. Visual assessment is not considered here as a “gold standard” but only as a reference to what exists in the best possible conditions in order for clinicians to appropriate the tools for quantifying locomotor asymmetries compared to what they are used to seeing and concluding subjectively with the classical (even imperfect) methods. This is a key issue for “calibrating” these new quantitative tools and ensuring that the slightest asymmetry recorded by the machine is not wrongly considered to be an expression of lameness, particularly in the circle. However, in order to minimize the limits of subjective examination, the veterinarians chosen in this study were highly experienced and trained in the same clinic. These factors have been shown to improve agreement between vets [31,36]. Confounding specialists and unexperienced vets, a 69% agreement between veterinarians was reached on forelimb lameness detection on circles [31]. Moreover, the lowest grade of lameness (1/10) was excluded because of weaker agreement for very subtle asymmetries [38], and sound horses were included based on the agreement of three veterinary specialists in order to decrease uncertainty [34]. The strict selection criterion resulted in a sample that only represents a proportion of lameness cases. This choice was made in order to study unambiguous and simple lameness cases as a first step.

This study aims to investigate spontaneous cases of lameness of various origins. However, it is obvious that lameness can have different clinical expression depending on the type of injury [1,39]. Circles may further reveal lameness unseen on the straight line [1,2,26]. Some indices may be modified according to the type of injury. More horses with specific types of injuries should be included to go further in this direction.

Other limitations induced by circles were the radius, speed and body lean angle. The diameter of the circle was imposed with a standardized examination area. However, these factors are difficult to control precisely in practice. Yet, they have been shown to affect the symmetry [9,24], although the relationship between asymmetry and body lean angle has been considered unpredictable [40].

## 5. Conclusions

This study established preliminary thresholds for the clinical interpretation of asymmetry indices when the horses lunged at trot on short circles under clinical examination conditions for simple limb lameness shown on both reins with a grade over 1/10. For discriminating sound horses from lame horses with the lame forelimb on the inside of the circle, the asymmetry thresholds were approximatively (average between right and left absolute values) 24% and 49% for, respectively, the minimum and the amplitude of the elevation of the head, and 12% for the temporal phase shift of the head. For discriminating sound horses from lame horses with the lame forelimb on the outside of the circle, the asymmetry thresholds were approximately (average between right and left absolute values) 10% and 9% for, respectively, the minimum and amplitude of elevation of the withers, and 2.5% for the temporal phase shift of the withers.

The results confirmed that circles on a hard surface induce the asymmetry of movements in sound horses. This asymmetry should not be confused with lameness, making the notion of threshold particularly important in this circumstance of the locomotion. In this study, asymmetry of the head movements was shown to be more effective to discriminate forelimb lameness when the lame limb was on the inside of the circle (sensitivity ≥ 69% and specificity ≥ 81%), whereas asymmetry of the withers movements was shown to be more effective to discriminate lameness when the lame limb was on the outside of the circle (sensitivity ≥ 72% and specificity ≥ 77%). The temporal asymmetry between the left and right duration of the vertical displacement cycle of the head or of the withers showed a discriminative power as high as asymmetry indices calculated from the amplitude of the vertical displacements. These quantitative results are useful in objectively helping clinicians to establish an informed diagnosis. In the future, these thresholds will be refined by including more horses and will be specific to the anatomical location and to the type of lesion responsible for the lameness. Indeed, this study needs to be extended by adding horses with different lameness types and horses that show lameness on only one rein. Also, horses with subtle lameness (1/10 grade) have not been included here, considering that the level of agreement between clinicians for very subtle asymmetries is low [23]. However, further studies will be needed to refine thresholds for those more subtle and uncertain cases. In addition, the investigation should be extended to the detection of hindlimb lameness and to circles on a soft surface.

## Figures and Tables

**Figure 1 animals-13-03319-f001:**
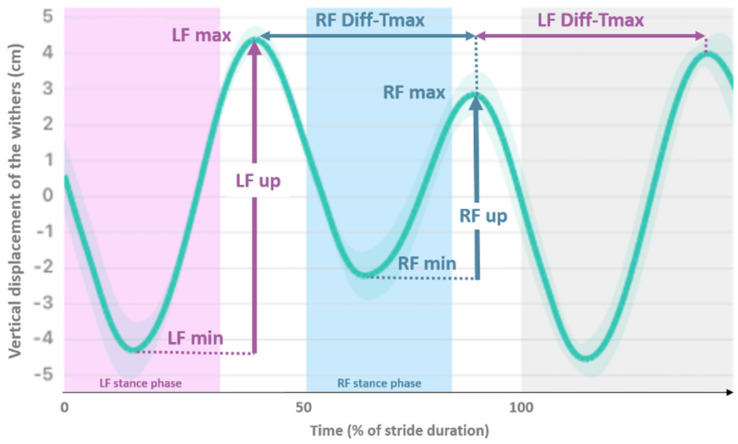
Mean vertical displacement (cm) of the withers plotted against time (expressed as percentage of stride duration) of a horse showing right forelimb (RF) lameness. Asymmetry Indices (AIs) are AI-min = RFmin−LFmin/LFup; AI-max = LFmax−RFmax/LFup; AI-up = LFup−RFup/LFup; and AI-Tmax = LFdiffTmax−RFdiffTmax/LFdiffTmax. (LF—Left Forelimb).

**Figure 2 animals-13-03319-f002:**
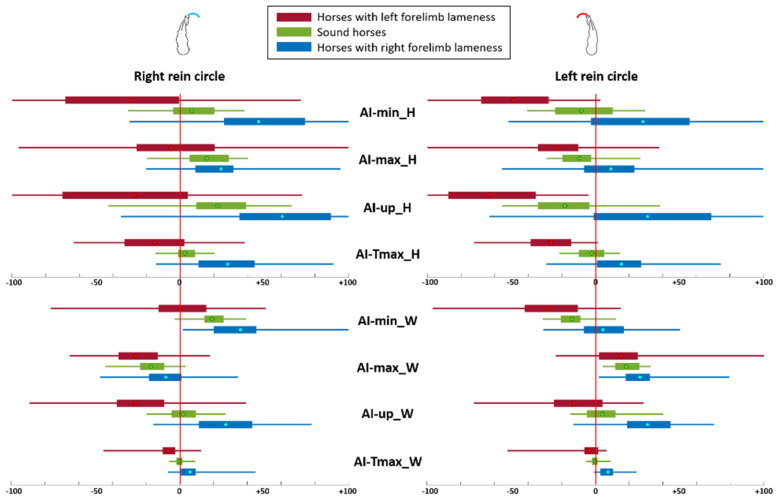
Boxplots of Asymmetry Indices (AIs) in percentage for sound horses and horses with a left forelimb lameness and right forelimb lameness. The dot represents the mean, the two extremities of the box are 25th and 75th percentiles, and extremities of the whiskers represent the minimum and maximum. A negative value reflects reduced movement or duration during the left limb stance, and positive value reflects reduced movement or duration during the right limb stance.

**Figure 3 animals-13-03319-f003:**
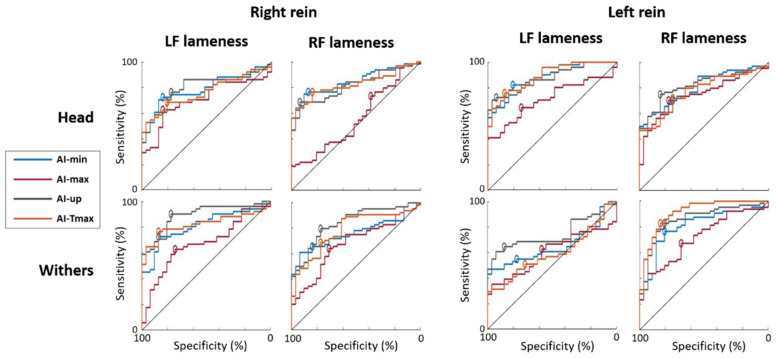
Receiver Operating Characteristic (ROC) curves discriminating horses with a constant Left Forelimb (LF) lameness from sound horses; and discriminating horses with a constant Right Forelimb (RF) lameness from sound horses, plotted for both reins on a circle. Tested Asymmetry Indices were AI-min (blue), AI-max (red), AI-up (grey) and AI-Tmax (orange). The highest specificity and sensitivity point of each curve is represented by a circle (top-left method). The black line is the hypothesized ROC curve with discriminative power only due to chance.

**Figure 4 animals-13-03319-f004:**
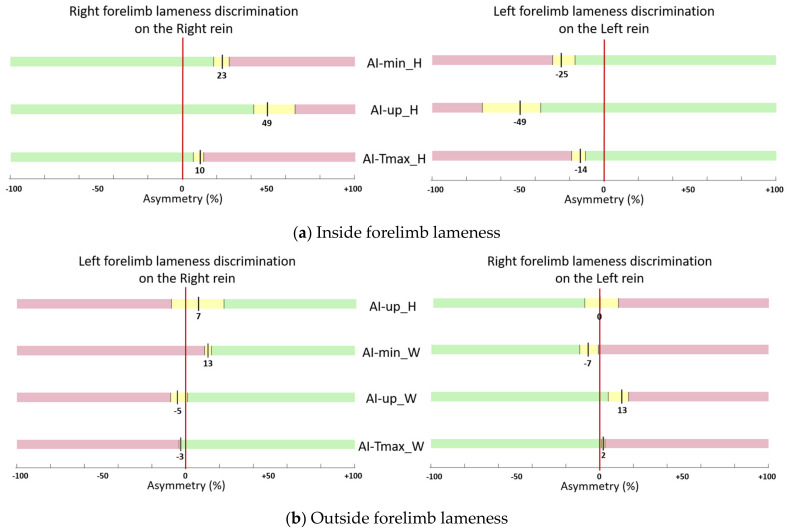
Thresholds (black line) of Asymmetry Indices (AIs) (in percentage of asymmetry) for lameness discrimination when the lame limb was on the inside of the circle (**a**) and when the lame limb was on the outside of the circle (**b**). Only the AIs with the sum of sensitivity and specificity over 150% for both right and left reins are plotted. Three ranges of values are represented: yellow for 95% confidence interval (95% CI) around the threshold, green for values below the 95% CI (“sound” horses) and red for values beyond the 95% CI (“lame” horses).

**Table 1 animals-13-03319-t001:** Number of horses showing lameness depending on the affected limb and the grade according to the 0–10 grade UK lameness scale. Mean ± SD ^1^ of lameness grade in each lame horse group.

Horses	Rein	2/10	3/10	4/10	5/10	6/10	7/10	Total	Mean ± SD ^1^
Left forelimb lameness	Left rein (inside lameness)	18	8	23	3	4	1	57	3.5 ± 1.4
	Right rein (outside lameness)	30	10	10	1	5	1	57	3.0 ± 1.4
Right forelimb lameness	Left rein (outside lameness)	28	13	23	3	3	0	70	3.1 ± 1.2
	Right rein (inside lameness)	23	14	19	5	8	1	70	3.5 ± 1.4

SD ^1^—Standard Deviation.

**Table 2 animals-13-03319-t002:** Mean ± SD of asymmetry indices (AIs) of the head and the withers in sound, left forelimb (LF) lame, right forelimb (RF) lame horses trotting on right rein and on left rein circles.

		Right Rein Circle			Left Rein Circle		
Location	AI	LF	Sound	RF	LF	Sound	RF
Head (_H)	AI-min_H (%)	−29 ± 39	7 ± 18	47 ± 32	−49 ± 26	−8 ± 20	28 ± 36
	AI-max_H (%)	−6 ± 34	16 ± 17	24 ± 23	−22 ± 28	−10 ± 14	9 ± 27
	AI-up_H (%)	−27 ± 48	22 ± 24	61 ± 32	−63 ± 28	−18 ± 25	31 ± 44
	AI-Tmax_H (%)	−15 ± 23	3 ± 9	28 ± 25	−26 ± 17	−2 ± 10	15 ± 21
Withers (_W)	AI-min_W (%)	0 ± 22	19 ± 10	36 ± 20	−28 ±24	−14 ± 11	4 ± 18
	AI-max_W (%)	−26 ± 16	−17 ± 12	−8 ± 16	15 ± 22	18 ± 8	26 ± 13
	AI-up_W (%)	−26 ± 25	2 ± 12	27 ± 22	−13 ± 21	4 ± 14	31 ± 19
	AI-Tmax_W (%)	−7 ± 9	0 ± 3	6 ± 9	−4 ± 9	0 ± 3	7 ± 6

**Table 3 animals-13-03319-t003:** Area Under the Curve (AUC) of the Receiver Operating Characteristic (ROC) curve discriminating forelimb lameness when the lame limb was on the inside of the circle from sound horses on circles, for each of the Asymmetry Indices (AIs). The highest sensitivity (Se), specificity (Sp) and associated threshold (Threshold) were calculated using the top-left method of ROC analysis. The 95% confidence intervals [;] were calculated by using a Bootstrap method, plotting ROC analysis on 400 population resamplings. Results for which the sum of sensitivity and specificity is over 150% for both circles (left and right reins) are in bold.

AI	Left Forelimb Lameness on the Left Rein (Inside Lameness on the Left Rein)				Right Forelimb Lameness on the Right Rein (Inside Lameness on the Right Rein)			
	AUC	Se	Sp	Threshold	AUC	Se	Sp	Threshold
**AI-min_H (%)**	**89 [83;94]**	**82 [72;94]**	**81 [65;90]**	**−25 [−30;−17]**	**86 [78;92]**	**79 [70;89]**	**87 [75;96]**	**23 [18;27]**
AI-max_H (%)	70 [60;80]	63 [47;76]	74 [57;89]	−17 [−25;−10]	57 [45;69]	74 [66;100]	39 [2;39]	9 [−11;12]
**AI-up_H (%)**	**87 [80;93]**	**72 [56;80]**	**94 [82;100]**	**−49 [−71;−37]**	**83 [74;90]**	**69 [54;78]**	**94 [84;100]**	**49 [41;65]**
**AI-Tmax_H (%)**	**90 [83;96]**	**79 [66;88]**	**87 [76;98]**	**−14 [−19;−11]**	**84 [76;91]**	**79 [70;90]**	**84 [71;95]**	**10 [6;12]**
AI-min_W (%)	66 [54;76]	53 [37;61]	87 [74;100]	−24 [−33;−19]	76 [67;86]	66 [52;77]	84 [69;97]	27 [24;32]
AI-max_W (%)	60 [49;73]	61 [46;83]	58 [23;71]	17 [11;25]	67 [56;77]	64 [48;74]	71 [56;88]	−11 [−15;−4]
AI-up_W (%)	76 [65;85]	67 [54;79]	81 [63;88]	−8 [−12;−3]	84 [76;91]	80 [69;92]	77 [62;88]	10 [4;14]
AI-Tmax_W (%)	61 [49;71]	60 [49;78]	61 [33;71]	−1 [−2;1]	79 [67;88]	67 [36;76]	77 [66;99]	2 [1;5]

**Table 4 animals-13-03319-t004:** Area Under the Curve (AUC) of the Receiver Operating Characteristic (ROC) curve discriminating forelimb lameness when the lame limb was on the outside of the circle from sound horses on circles, for each of the Asymmetry Indices (AIs). The highest sensitivity (Se), specificity (Sp) and associated threshold (Threshold) were calculated using the top-left method of ROC analysis. The 95% confidence intervals [;] were calculated by using a Bootstrap method, plotting ROC analysis on 400 population resamplings. Results for which the sum of sensitivity and specificity is over 150% for both circles (left and right reins) are in bold.

AI	Left Forelimb Lameness on the Right Rein (Outside Lameness on the Right Rein)				Right Forelimb Lameness on the Left Rein (Outside Lameness on the Left Rein)			
	AUC	Se	Sp	Threshold	AUC	Se	Sp	Threshold
AI-min_H (%)	79 [69;88]	70 [57;79]	84 [71;95]	−8 [−17;−1]	80 [73;88]	66 [45;75]	81 [63;98]	13 [0;34]
AI-max_H (%)	72 [60;82]	65 [50;75]	84 [73;99]	1 [−9;6]	75 [65;83]	70 [57;81]	77 [64;91]	−2 [−7;6]
**AI-up_H (%)**	**81 [71;90]**	**77 [62;89]**	**77 [61;89]**	**7 [−9;22]**	**82 [74;89]**	**74 [62;81]**	**84 [73;97]**	**0 [−9;11]**
AI-Tmax_H (%)	75 [65;85]	67 [55;79]	81 [65;91]	−3 [−7;4]	77 [68;85]	71 [57;82]	74 [59;88]	3 [−2;7]
**AI-min_W (%)**	**79 [71;89]**	**72 [60;82]**	**87 [75;99]**	**13 [11;15]**	**81 [72;89]**	**76 [63;86]**	**81 [66;94]**	**−7 [−12;−1]**
AI-max_W (%)	68 [57;79]	63 [49;75]	74 [59;88]	−23 [−28;−19]	71 [60;82]	66 [50;81]	68 [48;83]	22 [17;27]
**AI-up_W (%)**	**88 [81;95]**	**89 [81;100]**	**77 [60;85]**	**−5 [−9;1]**	**86 [78;93]**	**83 [73;94]**	**81 [65;89]**	**13 [5;17]**
**AI-Tmax_W (%)**	**81 [73;90]**	**74 [61;83]**	**87 [76;99]**	**−3 [−4;−3]**	**91 [86;97]**	**84 [72;93]**	**84 [71;94]**	**2 [1;3]**

## Data Availability

The data that support the findings of this study are available from the corresponding author upon reasonable request.

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
