# Peer review of "Asymmetry Thresholds Reflecting the Visual Assessment of Forelimb Lameness on Circles on a Hard Surface"

_animals, 2023, doi:10.3390/ani13213319_

Round 1

Reviewer 1 Report

Review of “Asymmetry Thresholds Reflecting the Visual Assessment of 2 Forelimb Lameness on Circles on a Hard Surface”

General:

Interesting manuscript presenting data aiming to introduce/clarify asymmetry thresholds/patterns for unilaterally forelimb lame horses.

Overall, there is one point that is not particularly clear (at least not to me):
According to the materials and methods section, all of the lame horses included in the study showed lameness on both reins (“Among them, 95 horses showed left forelimb lameness (LF) and 122 horses showed right forelimb lameness (RF) on both reins”). In essence, this is a very specific subset of horses that may or may not be very representative of horses seen in clinical practice which will (in my experience) more often than not show different types of lameness under different exercise conditions.

The specific selection of unilaterally, consistently lame horses across conditions fundamentally questions some of the analysis and presentation choices of the manuscript: is it possible to then differentiate between the categories used in the results section: ‘Discrimination between horses with internal forelimb lameness and sound horses’ and ‘Discrimination between horses with external forelimb lameness and sound horses’? The terms used would indicate to me, that these results are valid for the horses commonly seen in clinical practice, i.e. horses that are EITHER lame with the lame limb on the inside of the circle OR horses that are lame with the lame limb on the outside of the circle, while here the horses are lame on the same limb on both reins. So, do I understand correctly that both sections (‘Discrimination between horses with internal forelimb lameness and sound horses’ and ‘Discrimination between horses with external forelimb lameness and sound horses’) in fact report data from all horses, as opposed to one section dealing with inside forelimb lame horses and the other section with outside forelimb lame horses?

In addition, the specifically selected horses here, do not have concurrent hind limb lameness, see for example (Maliye et al., 2015) or (Rhodin et al., 2013), the latter for some interesting results on induced lameness and circle effects. The provided supplementary flow chart is very helpful in that respect: from 574 horses, only 217 had consistent forelimb lameness across straight-line and BOTH reins, and only 137 had consistent lameness >1/10. So a situation that is not particularly common in the overall population from which the cases were selected.

I totally understand the decision to exclude horses with 1/10th lameness out of the following reason(s): data analysis and calculation of thresholds between sound and lame will eliminate considerable ‘overlap’ between the two distributions (sound vs lame) likely related to some difficulties in agreeing on what is a grade 1 lameness (Keegan et al., 2010). The latter being the second reason. However, from a data analysis sense, removing this overlap (the grade 1 lame horses), likely influences the calculated sensitivity and as a result will likely lead to a shift in the calculated thresholds. While it appears unreasonable to reanalyze the data including 1/10 lame horses, this limitation is another reason for being very clear about the relevance of the presented results for a very specific group of horses.

The whole manuscript (including the summaries) could do with careful rewording (and restructuring) to make sure that it is very clear throughout that the results and conclusion are valid for the specific group of horses selected and investigated.

These are still some very interesting data ands in particular figure 2 is super helpful for illustrating how movement asymmetry values change between exercise conditions and the sensitivity/specificity/threshold results are interesting to document what is happening in a very specific group of horses with consistent forelimb lameness across straight-line and both lunge directions and no concurrent hind limb or contra-lateral forelimb lameness.

Further comments: 

It would also be very nice to provide a full data table (supplementary information) of these horses including the ‘full’ data sets, i.e. the movement symmetry parameters of head, withers and pelvis.

It would be very interesting to document the compensatory patterns (pelvic movement symmetry) in this specific group of horses! Consider including pelvic movement symmetry illustrations (for example in the style of figure 2). That would be super helpful.

Terminology:

Replace ‘internal’ and ‘external’ limb by ‘inside’ and ‘outside’ limb? Internal respectively external sounds like one is looking at internal structures and the other at external structures of the limb. Also, this needs to be revised with respect to the comments about the specific lameness type (consistent unilateral lameness across reins) and the question about the analysis of the differences between ‘inside limb’ and ‘outside limb’ lameness in that context when the horses are consistently lame across reins.

Abstract:

Currently the abstract is not presenting any quantitative information. What are the values of the ‘best thresholds’ for example and what are sensitivity and specificity of these thresholds.

Please be very specific about the specific group of horses for which this study has been conducted.

Introduction

“Indeed, circles induce specific forces on the limb and 3D joint movements comparing to the straight line, like higher loading force on the external limb, collateromotion and axial rotation [1,2]. These specific biomechanical constraints on the anatomical structures are considered to be responsible for a different symptomatology between the circle and the straight line [1]. ” Please check the sentence structure of the first sentence: do you mean ‘compared to’ ? Also: what about body lean angle and the need to produce centripetal force: are these not likely also associated with the different degrees of lameness that can be seen on circles? 

“Recently, asymmetry thresholds for relative indices have been determined on the straight line [14].” Might be worth mentioning the threshold values here?

“as a large proportion of the movements recorded on the circle must be considered as physiologically asymmetric.” I feel this sentence needs some references? They are following a few sentences later but might be good to have them here already?

Results:

“Hannover”? “Hannoveranian’?

“KWPN”: acronym needs to be explained before first use.

“The stride frequency was of 0.79 ± 0.05 stride per second on the both reins.” Stride frequency? Is this not more likely “stride duration’? 0.79 strides/s, so each stride had a duration of >1 second? This would likely be ‘walk’?

In general, it would be helpful to get an idea of the ‘range of motion’ that has been used to calculate the asymmetry indices. At the very least, the results tables should list the relevant range of motion values. Even better, the complete data set, i.e. all relevant values for each horse (all symmetry values and range of motion values) should be made available as supplementary material for example in a filesharing database so in future meta-analysis combining these data with other data bases would be possible?

Figure 2 caption: ‘Boxplots of Asymmetry Indice ..’: either ‘index’ (singular) or ‘indices’ (plural)? Otherwise super interesting figure ! Consider adding pelvic movement symmetry here to document any (if any?) compensatory movements.

“For discriminating internal forelimb lameness, the indices with the greater sensitivity and specificity (sum of sensitivity and specificity greater than 150%) were three indices from the head (AI-min_H, AI-up_H, and AI-Tmax_H), whether the lameness occurs on the left forelimb on the left rein or on the right forelimb on the right rein.” Please clarify this sentence: ‘discriminating forelimb lameness’? From what?

Discussion: see ‘general comments’ and devise a clear strategy to clarify how the differentiation between inside and outside lameness makes sense here for the specific group of horses sowing consistent lameness on both reins.

Conclusion: include some numerical values for the most important findings.

Reviewer 2 Report

Overall

This manuscript evaluates the asymmetric gait in horses presenting a lameness on the lunge at the trot travelling on both reins.

Fundamentally, this study holds merit, however grammatically there are significant issues throughout and the continuance or the flow of the topic in the study becomes confusing as a result.  This becomes disconcerting for the reader trying to connect the points from one topic to the next.

Inconsistency with acronym use throughout the study, e.g., ROC (finally described Line 135) or IMU in Key Words not explained (?). Each needs to be established or explained such as what is ‘the CIRALE’?

The repetitive use of discriminating, discriminative, best and expert are words that need to be reconsidered in this manuscript.

Attached the Supplementary File as an example of grammar changes.

Summary

Requirement 200 words – 55 still available.

This is a section in the study that requires simpler explanations for the ‘layperson’. The first four sentences require further elaboration for clarity and describe Asymmetry Indices (AI) here.   

Abstract

Requirement 200 words – 0 available.

Good introduction to the topic. Acronym already established for Asymmetry Indices (in Summary and Abstract which is correct) and should only require AI thereafter but spelt fully at the beginning of a sentence.  

Interesting use of words and recommend changes more in keeping with the vernacular – ‘best discriminative power’; ‘minima and maxima’ (rarely used in equine literature); ‘expert’ (suggest specialist); ‘discriminate’ (suggest differentiate/ing, classify or similar).

Introduction

Excellent points are made here, yet the same points regarding grammar, and word use apply.

Materials and Methods

Further explanation relating to ‘the CIRALE’ is required.

Previous points regarding grammar, and word use apply. Need to reference the use of S1 (Supplementary file) here.

Suggest explanation of the trot as to WHY this gait is preferred above others, e.g., at the trot the head of the horse is still during the 2 beat diagonal stride whether on straight line or circling to left or right in a sound horse. 

Line 91 – ‘hard surface’ – concrete? In line 86 you described the ground surface as rubber pavers, but what was the hard surface composed of?

Results

Points regarding grammar, and word use apply.

Presumption that KWPN is a known acronym. 

Read Table and Figure comments below.

Tables

Table 1. Location – do you mean limb?

A key is required for Tables. 2,3 and 4.

Figure

Figures. 1,2,3 and 4. The writing and numbers are difficult to read. Spell out percentage % (Fig. 2)

Discussion and Conclusion

Some grammar issues. Limitations well defined, however, as the Figures were unreadable, comments for these two sections are limited mostly to the 2nd round.

References

Inconsistencies in referencing formats – incomplete references, abbreviation of journals, authors names, page numbers for referenced material from books.  

See previous notes and file.

Reviewer 3 Report

This is a great paper with statistically significant findings.

It would be good to have a picture or diagram to show the locations of the sensor for measuring head & wither movements measured by the EUISYM. 

The lameness grade results in Table 1 may require a normality test as the distribution of the data did not appear to be in normal distribution. Would the Median value with range be more appropriate to express the results ?

Thresholds for lameness detection have been defined for the trot on a straight line, but are not applicable to the circles. This study aims to select the asymmetry indices with good discriminant capacity between sound and forelimb lame horses on a hard surface circle and then to define the threshold value associated.

This study established thresholds for the clinical interpretation of asymmetry indices when the horses are lunged at trot on short circles under clinical examination condition. The results confirmed that circles on a hard surface induce asymmetry of movements in sound horses. This asymmetry should not be confused with lameness, making the notion of threshold particularly important in this circumstance of the locomotion. In this study, asymmetry of the head movements was shown to be more effective to discriminate lameness affecting the limb inside the circle, whereas asymmetry of the wither movements was shown to be more effective to discriminate lameness affecting the limb outside the circle.

Given that the distribution of the lameness score did not appear to be in a normal distribution. Normality testing is advised. MEDIAN (with range) and standard deviation (SD) can be calculated from data for sound horses and for horses showing lameness on the internal and on the external forelimb, collected on each rein. Non-parametric statistical analysis can be considered for comparison of lameness score for inter-rater assessment

The conclusions are consistent with the evidence and arguments presented
and they address the main question posed.

The references are appropriate.

Round 2

Reviewer 1 Report

I sincerely thank the authors for reworking the manuscript according to the suggestions made. It is a great contribution to the literature! Thank you.

minor edits:
replace 'both the reins' with 'both reins' (in a few places)

replace 'guaranty' with 'guarantee' 
